# Modeling the Real World with High-Density Visual Particle Dynamics

**William F. Whitney**[*]  **Jacob Varley**[*]  **Deepali Jain**[*]  **Krzysztof Choromanski**[*]
Google DeepMind   Google DeepMind   Google DeepMind   Google DeepMind

**Sumeet Singh**          **Vikas Sindhwani**
Google DeepMind          Google DeepMind

**Abstract:** We present *High-Density Visual Particle Dynamics* (HD-VPD), a learned world model that can emulate the physical dynamics of real scenes by processing massive latent point clouds containing 100K+ particles. To enable efficiency at this scale, we introduce a novel family of Point Cloud Transformers (PCTs) called *Interlacers* leveraging intertwined linear-attention Performer layers and graph-based neighbour attention layers. We demonstrate the capabilities of HD-VPD by modeling the dynamics of high degree-of-freedom bi-manual robots with two RGB-D cameras. Compared to the previous graph neural network approach, our Interlacer dynamics is twice as fast with the same prediction quality, and can achieve higher quality using $4\times$ as many particles. We illustrate how HD-VPD can evaluate motion plan quality with robotic box pushing and can grasping tasks. See videos and particle dynamics rendered by HD-VPD at https://sites.google.com/view/hd-vpd.

**Keywords:** point clouds, particle dynamics, world models for control, Performers

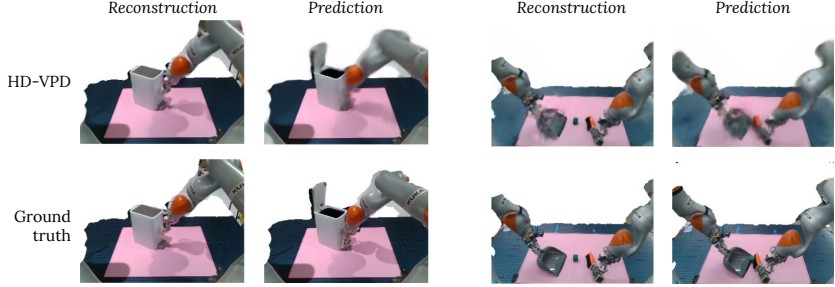

Figure 1: HD-VPD can accurately predict the dynamics of complex real-world interactions between robots (here, 16-DoF bi-manual Kukas) and objects/tools. **Left:** a push-pedal trash-can opening task and **Right:** a bimanual dustpan sweeping task. **Top row:** Renders from HD-VPD. The first image in each pair is a reconstruction of the matching input frame, and the second is a prediction several timesteps into the future given a sequence of robot actions. **Bottom row:** Ground-truth test set video frames.

## 1 Introduction

Physical simulators are the linchpin of modern robotics, enabling cheap data collection, safe verification of control algorithms, and real-time control via planning. Traditional analytic simulators [1, 2, 3, 4, 5] are fast and convenient to use, but lack the ability to precisely match the complex objects and dynamics of real-world scenes. Learned dynamics models make fewer assumptions on the form of objects and their interactions, but typically come with onerous constraints on their training data, e.g., requiring information such as 3D meshes and poses for all objects or per-object segmentation masks [6, 7, 8].

---

[*]Equal contribution. Correspondence to wwhitney@google.com.

8th Conference on Robot Learning (CoRL 2024), Munich, Germany.

To address this problem, new classes of learned dynamics models have been proposed that can train directly on multi-view RGB-D observations, with no object-centrality in their data requirements or representations. A state-of-the-art approach of this flavor is Visual Particle Dynamics (VPD) [9], which represents scenes as a collection of 3D particles whose interaction dynamics is governed by graph neural networks (GNNs) [10, 11] and rendered to images with a conditional NeRF [12]. VPD models are trained end-to-end with a video prediction loss, support 3D state editing and multi-material dynamics modeling, and are data-efficient enough to model simple dynamic scenes with $< 50$ training trajectories. However, VPD has only been applied to simple simulated scenes under passive dynamics (without actuation), and it is unable to scale beyond $\sim$30K particles due to memory and speed limitations. For applications in robotics, VPD needs two extensions: (1) to take robot actions into account, and (2) to be able to model environments with a much higher level of detail. This is the focus of this paper.

We propose a *High-Density Visual Particle Dynamics* (HD-VPD) world model which can train on robot interactions in real scenes and model their dynamics with 100K+ particles. To reach this scale, we propose a new class of Point Cloud Transformers (PCTs) [13] called *Interlacers*, which intertwine linear-attention Performer [14] layers and local neighborhood attention layers. We demonstrate the capabilities of HD-VPD by modeling the dynamics of bi-manual Kuka robots with multi-view depth input data. We find that Interlacer dynamics models are able to equal GNN's prediction quality in half the time using matched point cloud densities, and that they can exceed the best GNN's quality by leveraging high-density point clouds with $4\times$ as many particles as the GNN can fit in memory. We complement these results with illustrative experiments using HD-VPD for downstream applications such as planning in manipulation, leveraging HD-VPD's ability to capture a scene from a single observation and define goals in 3D space. See videos and particle dynamics predictions from our model at https://sites.google.com/view/hd-vpd.

Our main contributions are as follows:

1. We present High-Density Visual Particle Dynamics (HD-VPD) world models that train end-to-end on real robotics data. They take RGB-D images and robot kinematic skeletons representing actions as inputs, and they predict future 3D particles and rendered images.

2. To enable HD-VPD to operate on large, detailed scenes, we propose a new class of Transformers for point clouds called *Interlacers*, which scale linearly (vs. quadratically) with point cloud size. Interlacers combine Performer-PCT layers [14, 15] and memory-efficient GNN-inspired local attention layers, enabling efficient global point-to-point attention (with the former) while maintaining high-fidelity local geometric detail (with the latter).

3. We show that HD-VPD can deliver realistic action-conditioned video predictions on complex scenes of bi-manual robots interacting with various objects. We find that the Interlacer architecture enables fast, memory-efficient, and high-fidelity video prediction with HD-VPD. Whereas typical NeRF models require tens or hundreds of camera views for training, HD-VPD uses only two, making real-world data collection straightforward. We use these models to illustrate downstream applications of HD-VPD in robotic bi-manual control, where the HD-VPD world model is able to predict the success or failure of candidate plans.

## 2 HD-VPD: High-density visual particle dynamics world models

HD-VPD is a 3D dynamics model disguised as a video prediction model. It takes RGB-D images and robot actions as inputs and predicts images of the future, just like a video world model. Under the hood, though, it represents both the current state of the world and the actions which operate on it as particles in a shared 3D space. To make a prediction several timesteps into the future, HD-VPD goes through three steps, shown in Figure 2:

1. Encode input images from all views and times into 3D particles that represent the state of the scene.

2. Predict the dynamics of the scene. Combine particles representing the first robot action with those representing the scene state and predict the change in state. Use this to update the state. Then repeat this procedure for the next robot action, and so forth.

3. Render the particles representing the predicted state of the scene into an image.

In this way, HD-VPD can make predictions many steps into the future while remaining in particle space, only rendering to pixel space as needed for training or visualization. The rest of this section describes the HD-VPD model, architectural choices, and training in more detail.

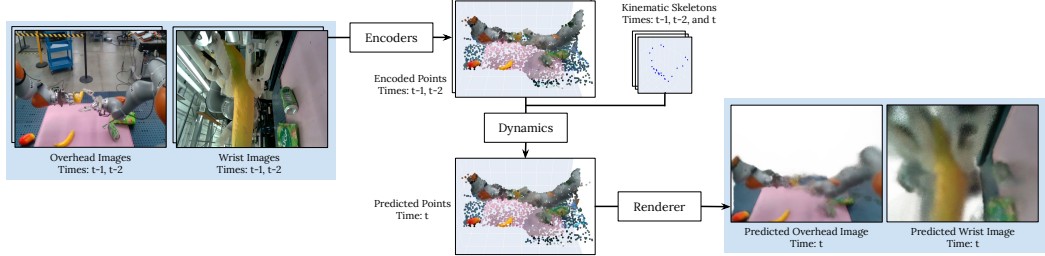

Figure 2: Overview of HD-VPD model with learned Encoders, Dynamics and Render. **Encoders** encode RGB-D images into a point cloud representation with latent per-point features. **Dynamics** predicts the evolution of the scene conditioned on the current scene as well as a kinematic skeleton representing the motion of the robot. **Renderer** is a Point-NeRF style model which enables generation of images of the predicted future scene. The entire model is trained end-to-end with a pixel-wise $L^2$ loss.

**Encoding**   HD-VPD receives RGB-D images from multiple cameras (in our experiments, 2) at timesteps $T - l, \ldots, T - 1$ for an input window of length $l$; throughout this work we use $l = 2$. Each RGB image is first encoded into feature image of per-pixel feature vectors using a U-Net [16]. Using known camera intrinsics and extrinsics, combined with the depth channel of the RGB-D input, each of these per-pixel features can be unprojected into the global coordinate frame, forming a point cloud where each point corresponds to an input pixel. Each point is associated with the predicted feature vector from its corresponding pixel to form a particle represented as a (location, feature) tuple $(\mathbf{x}, \mathbf{f})$. The particles from across cameras are merged together within one timestep before being subsampled uniformly at random to a fixed total number of particles $N$.

**Action representation**   Unlike VPD, HD-VPD includes a representation of actions, allowing it to model robotic scenes and be used for planning and other forms of controllable generation. We represent actions as a set of *kinematic particles* describing the motion of the robot across multiple timesteps: previous steps $T - 2$ and $T - 1$, and next step $T$ which represents where the robot plans to move. The particles from each timestep are located at the joints, fingertips, and the base of the grippers of the two arms. Each kinematic particle is associated with a feature vector consisting of two concatenated one-hot vectors: one indicating which robot joint it is, and one indicating which timestep it came from. By representing actions where they occur in 3D space, HD-VPD can learn to associate them directly to the scene particles which they affect. More details and an example are in Appendix C.

**Dynamics**   We employ the Interlacer architecture, described in detail in Section 3, for the dynamics. To predict the scene state at time $T$, the dynamics model takes as inputs the scene particles from the encoder corresponding to timesteps $T - 2$ and $T - 1$ and the kinematic particles corresponding to timesteps $T - 2$, $T - 1$, and $T$. The Interlacer encodes each timestep's scene particles and the full set of kinematic particles separately before combining them in one large trunk. Predictions are made in the form $(\Delta \mathbf{x}_i, \Delta \mathbf{f}_i)$ for each particle $i$ from timestep $T - 1$. The prediction for timestep $T$ can then be constructed as $\{(\mathbf{x}_i^{T-1} + \Delta \mathbf{x}_i, \mathbf{f}_i^{T-1} + \Delta \mathbf{f}_i)\}_{i=1}^N$.

**Rendering**   The renderer uses a ray-based renderer conditioned on a point cloud, similar to Point-NeRF [17], to render images. This involves casting rays through the scene and, at each sampled location along the ray, finding a set of neighboring particles. Summary statistics describing these particles are computed and then provided as input to a NeRF MLP, which predicts the color and density of that location in the scene. These predictions are composited along the ray to produce a rendered pixel. For more details, refer to [9].

**Training**   During training, we encode a set of input RGB-D images into point clouds, then recursively apply the dynamics model with actions from time $T \ldots T + K$ to make predictions multiple timesteps into the future. For supervision we sample a small set of rays to render at each timestep and compute the $L^2$ loss between the predicted and observed RGB values. Unlike typical NeRF models, which require tens or hundreds of views for training, we train HD-VPD with data from just

two cameras at each timestep. Both views are provided as input to the model, and the training loss is computed on pixels sampled from both views.

# 3 Interlacers: when linear-attention Transformers and GNNs meet

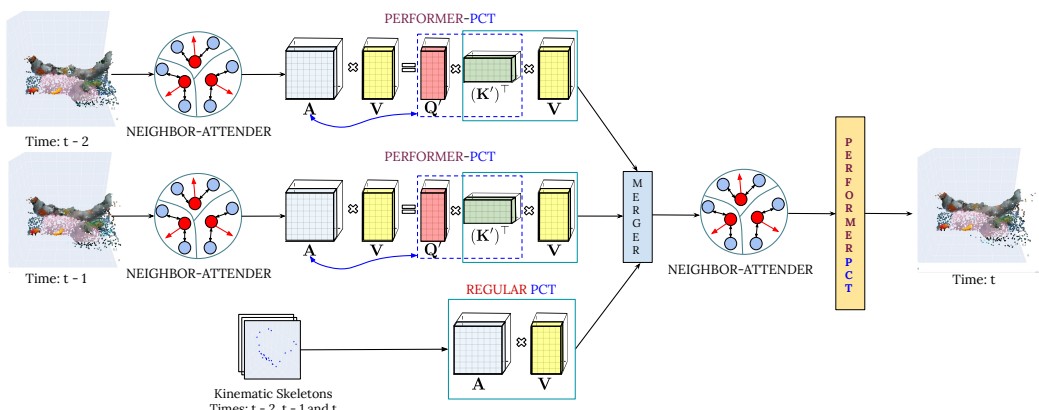

Figure 3: **Interlacer dynamics**. The input point clouds from each timestep are processed by the neighbor-attender layers, followed by the Performer layers (see Appendix A for details). In the HD-VPD model, a separate third channel is reserved for processing kinematic particles describing the actions conducted by the robot, which are preprocessed by a regular PCT layer. Then, all of the preprocessed point clouds are merged. The model predicts particles' displacements as well as deltas of their corresponding feature vectors, after applying one more neighbor-attender and Performer. See Figure 2 for how Interlacer is integrated with the HD-VPD model.

To make a prediction for time $T$, the Interlacer takes as input the scene state from each timestep in a given window, where each scene state $\mathcal{S}_t = \{\mathcal{X}_t, \mathcal{F}_t\}$ ($t = T - l, ..., T - 1$ for the window length $l$) consists of the set of the 3D positions $\mathcal{X}_t = \{\mathbf{x}_1^t, ..., \mathbf{x}_N^t\} \in \mathbb{R}^3$ of a given set of particles $\mathbf{X}$ in time $t$ and their corresponding feature vectors at that time: $\mathcal{F}_t = \{\mathbf{f}_1^t, ..., \mathbf{f}_N^t\} \in \mathbb{R}^f$ (for some $f > 0$). For modeling robotics data, the Interlacer also receives kinematic particles at timesteps $T - l, ..., T$ representing the action the robot will take.

The Interlacer consists of two main types of layers: (1) linear attention Performer-PCT layer [14, 15] and (2) graph-based feature-agglomeration layer, leveraging structural inductive priors encoded by local neighborhoods in graphs, that we refer to as *Neighbor-Attender*. The input point cloud for each timestep is first processed by the Neighbor-Attender modules. Their outputs are then processed by Performer-PCT modules, which produce versions of their input point clouds with updated features. These feature point clouds from each timestep are then merged together into one large point cloud along with the robot's skeleton points from all times. The Interlacer then applies one more Neighbor-Attender layer, followed by a final block of Performer-PCT layers. This model predicts the change in location and features $(\Delta \mathbf{x}_i, \Delta \mathbf{f}_i)$ for each particle $i$ in the last input timestep $T - 1$. These deltas are applied to $(\mathbf{x}_i^{T-1}, \mathbf{f}_i^{T-1})$ to construct a particle prediction at time $T$, which is fed back into the dynamics to predict another step forward or into the renderer for image generation.

The Performer-PCT [14, 15] is described in detail in Appendix A. Next, we discuss *Neighbor-Attender*, a novel mechanism to incorporate geometry into attention.

## 3.1 Incorporating geometry into attention: the Neighbor-Attender layer

The Neighbor-Attender, illustrated in Figure 4 is designed to provide each particle with information about the geometry and features of its immediate neighborhood as efficiently as possible. A simple approach might find the $k$ nearest neighbors of each particle and extract a summary of those neighbors' features and relative positions, but this would involve a forward pass on $kN$ particle-neighbor pairs. Inspired by RandLA-Net [18], Neighbor-Attender instead computes such neighborhood features only on a small, uniformly-sampled *subset* of particles we call *anchor particles*, then uses the anchor particles to update the rest of the particles. This results in a bottleneck step of only $\frac{kN}{r}$ pairs

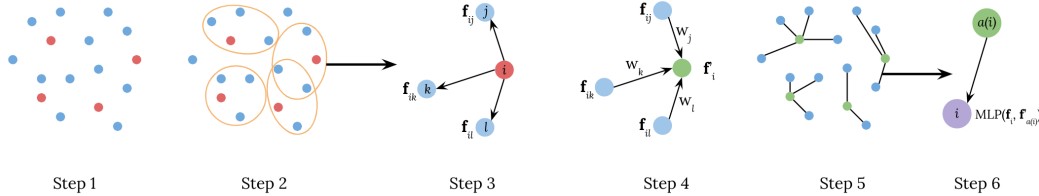

Step 1    Step 2    Step 3    Step 4    Step 5    Step 6

Figure 4: The anatomy of the Neighbor-Attender layer. A subset of anchor particles are sampled uniformly from the full set of particles, and these anchor particles aggregate information from their $k$ nearest neighbors. Then the full set of particles are updated, each using the features of the closest anchor particle.

for a subsampling rate $r$, allowing us to control memory consumption at will. The computations of the Neighbor-Attender layer consist of six steps that we explain in detail below; steps 1-4 aggregate neighborhood features onto the anchor particles, then steps 5-6 use those to update the rest.

1. **Choosing anchor particles:** Sample uniformly at random $r = \frac{N}{4}$ particles from the input set of $N$ particles. We refer to them as *anchor particles*.

2. **Computing neighbors of anchors:** For each anchor particle $i$, compute the set $\tau_i$ of its $s = 16$ nearest neighbors from the entire $N$-element set.

3. **Computing attention-vectors:** For each anchor particle $i$ and its neighbor $j \in \tau_i$, compute the relative position feature vector defined as: $(\mathbf{x}_i, \mathbf{x}_j, \mathbf{x}_i - \mathbf{x}_j, \|\mathbf{x}_i - \mathbf{x}_j\|_2)$ and concatenate it with the feature vector $\mathbf{f}_j$. We refer to the resulting vector as the $(i, j)$ *edge feature* $\mathbf{f}_{ij}$.

4. **Updating feature vectors for all anchor particles:** For each anchor particle $i$, compute its new feature vector $\mathbf{f}'_i$ as the weighted sum of its edge features using a learnable MLP module:
$$\mathbf{f}'_i = \sum_{j \in \tau_i} \frac{\exp\left(\mathrm{MLP}(\mathbf{f}_{ij}; \theta_1)\right)}{\sum_{k \in \tau_i} \exp\left(\mathrm{MLP}(\mathbf{f}_{ik}; \theta_1)\right)} \mathbf{f}_{ij} \tag{1}$$

5. **Finding closest anchors for all the particles):** For each particle $i$ in the original $N$-element set, find its closest anchor particle $a(i)$.

6. **Computing new feature vectors for all the points:** For each particle $i$ in the original $N$-element set, compute its new feature vector $\mathbf{f}''_i = \mathrm{MLP}(\mathbf{f}_i, \mathbf{f}'_{a(i)}; \theta_2)$.

## 4 Experiments

### 4.1 Datasets and HD-VPD instantiations

**Hardware and dataset** We train HD-VPD models on a dataset of bi-manual Kuka robots interacting with a variety of objects and tasks. The dataset consists of episodes collected from 5 different robots over several months. The robots (details in Appendix B, [19]) are equipped with an overhead RealSense and left arm wrist RealSense cameras capable of providing RGB-D images. The dataset is an uneven distribution of approximately 60 tasks. We break the episodes into snippets of trajectory of at most 8 steps with 500ms passing between consecutive steps. The dataset is 1.6TB.

**Models** We train HD-VPD models with three different dynamics architectures and varying numbers of particles for the experiments below. These architectures are: (1) a hierarchical **GNN** using the architecture from [9] with the addition of kinematic particles for actions; (2) **Performer-PCT**, which consists of repeated Performer-PCT layers; and (3) **Interlacer**, which includes both Performer-PCT and Neighbor-Attender layers. Architecture and training details are available in Appendix E.

### 4.2 Learned bi-manual robot dynamics with HD-VPD

In this section, we present learned dynamics obtained with our HD-VPD method for the bi-manual Kuka robot. Corresponding videos are available at `https://sites.google.com/view/hd-vpd`.

**Video quality scales with particles** We evaluate the quality of next-step image predictions (measured by SSIM) on the test set made with each architecture using varying numbers of particles. In

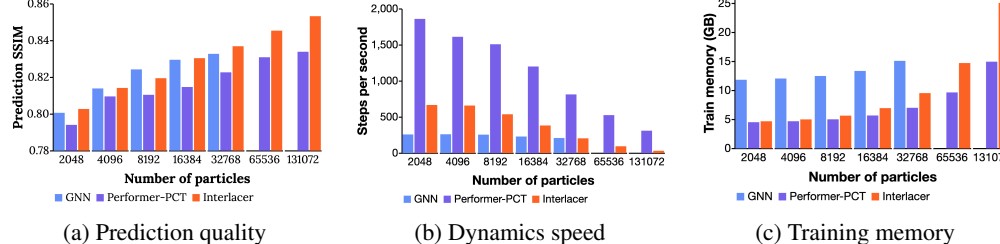

(a) Prediction quality      (b) Dynamics speed      (c) Training memory

Figure 5: Analysis of models' behavior as a function of number of particles. Note GNN is unable to be run with 65K or 131K particles due to memory limitations. Interlacer with 131K particles provides the best prediction quality while staying competitive with the GNN baselines for dynamics speed, and with Performer-PCTs for memory requirements. **(a)** Test set SSIM prediction quality increases with the number of particles, and Interlacer with 131K points does the best. **(b)** Interlacer is faster than GNN while able to handle many more particles. Performer-PCT is faster, but achieves worse results. **(c)** Performer-PCT and Interlacer use less memory than the GNN baseline, enabling them to scale to larger point clouds.

Fig. 5a, we show that the Interlacer-based HD-VPD model produces comparable-quality predictions to the GNN at every number of particles. The GNN model is unable to be trained at 65K and 130K particles due to memory consumption, whereas the Interlacer model continues to improve with scale. The Performer-PCT model lags behind the GNN and Interlacer at each number of particles. We also explore the behavior of these models on multi-step predictions, where the increasing uncertainty associated with long-range prediction might render high-density models' image quality advantage moot. Figure 6 shows that while Interlacers of all sizes are less accurate the further into the future they predict, using more particles continues to produce better predictions.

**Computational requirements** We evaluate the key computational costs associated with running and training on the three classes of models. Figure 5b shows the number of dynamics steps per second achieved with each architecture and number of particles. The Interlacer is at least as fast as the GNN model at each scale, in some cases providing more than 2x speedups. The Performer-PCT is the fastest by a significant margin, but it is unable to match the image quality of the other models. In terms of memory consumption (Figure 5c), the GNN model always requires considerably more memory to train than the Interlacer or Performer-PCT models, and is unfeasible to use at scales beyond 32K.

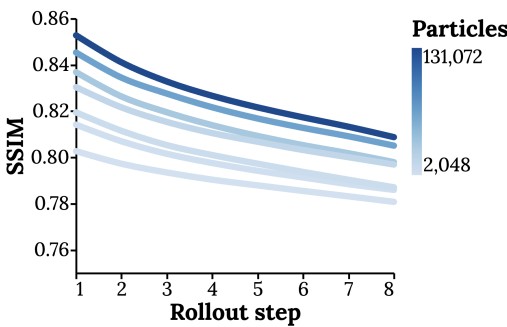

Figure 6: Video quality with longer rollouts and increasing particle density.

## 4.3 Downstream experiments on hardware

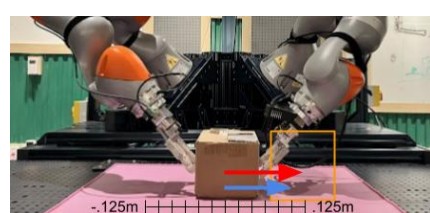 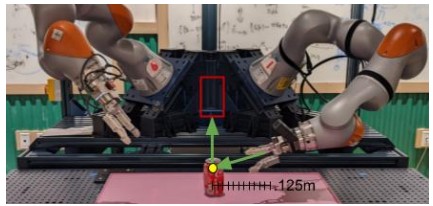

Figure 7: Experimental setups. **Left:** Box pushing task setup. We measure cost as the error between a goal push vector (red) and learned dynamics rollouts of candidate push vectors (blue). **Right:** Grasping task setup. We measure cost as a function of the distance upward travelled by the Coke can particles and compare against real world grasp success.

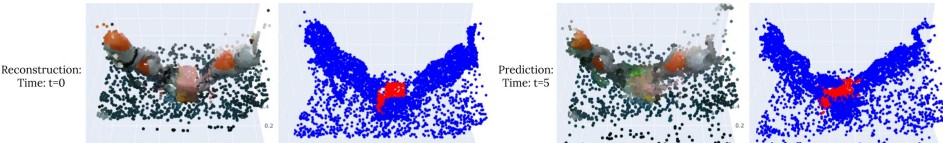

Figure 8: **Top:** Box Pushing Dynamics Rollout. HD-VPD prediction of scene dynamics from a left push motion. The particles on the box move in the direction consistent with the push.

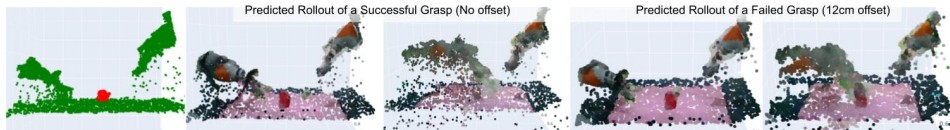

Figure 9: Grasping Dynamics Rollouts. **Left:** Red points belong to the can whose height is tracked. **Center:** shows the motion of a successful grasp (Coke can points merge with gripper and lift). **Right:** shows HD-VPD prediction for a failed grasp (Coke can points don't leave the table).

**Box push plan selection** This task is to move the box a 12.5 cm to the right as shown in Fig. 7. We have a fixed set of motion plans that move between 12.5 to -12.5 cm. These planned motions are rolled out with HD-VPD. By segmenting the box points from the input point cloud, we define the cost of a push plan as the median distance between the predicted and goal box points. Lower cost for a plan implies that the model predicts the box will be closer to the goal. Figure 10a shows that HD-VPD's predicted costs are close to ground truth (distance 0.25 for a large push the wrong way, 0.0 for a push to the right location). Choosing the minimum-cost plan under HD-VPD's cost predictions leads to optimal behavior in this experiment.

**Grasp plan selection** This task is to lift a Coke can straight up off the table to a position 0.2m in the air. We have a fixed set of planned grasps at various locations relative to the object, with offset 0.0 corresponding to grasping at the center of the object. We roll out HD-VPD to get predicted outcomes of each plan. By segmenting the points corresponding to the can out of the point cloud, we can define a cost function for each plan. Lower cost is better and implies can particles are closer to the goal. We show that the HD-VPD model cost increases as the planned grasp pose moves away from the object and becomes less likely to succeed as shown in Figure 10b. The HD-VPD model is able to infer that the Coke can points will be lifted on the grasp motions that do succeed in the real world (blue points). More details in Appendix D.

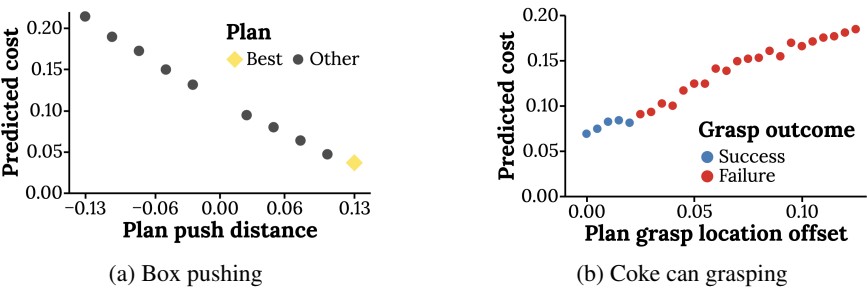

(a) Box pushing        (b) Coke can grasping

Figure 10: Results on two planning tasks. **(a)** Results for planning a desired 0.125m push. X-axis is planned push distance for different potential push plans, Y-axis is the cost for each plan as measured via the HD-VPD rollout. The yellow point shows that a planned push of 0.125m is the best option when the goal is to move the box 0.125m, and the cost of other push plans are ordered consistently by this cost function. **(b)** Grasp offset from object center vs HD-VPD predicted plan cost. With sufficiently large offsets, the grasps miss the can and fail. HD-VPD latent dynamics is able to understand that the points of the coke can should lift up following a grasp trajectory for well planned grasps, and understands that the can should not lift upwards for poorly planned grasps that fail in real world execution.

**Dataset inspection** After training HD-VPD, we run it on trajectories from the training set and find those with anomalously high loss. This surfaces previously-unnoticed errors and outliers in the data, including sessions with incorrect geometric camera calibration, faulty camera exposures, and humans in the robot workspace. Such dataset introspection may be useful when preparing a dataset for training models such as policies. Examples are shown in Appendix F.

# 5 Related work

**Dynamics models for robotics** Accurately modeling robot dynamics is essential for tasks such as motion planning, control, and simulation. Traditional approaches rely on rigid body dynamics models [1, 4, 2, 5, 3], but these models can be inaccurate in complex environments where objects are deformable or interacting in unpredictable ways, and require painstaking authoring of scenes and assets. A significant literature has explored the use of learned models for robot dynamics, which can broadly be divided into two camps: 2D and 3D. 2D models operate directly on pixels and use architectures designed for image generation [20, 21]. While these have found some success by scaling up to modern foundation model sizes and datasets [22, 23], their learned dynamics can be nonphysical, and they lack 3D representations that can be used for cost functions and scene editing.

More closely related to this work are the 3D learned dynamics models. Various works have used datasets of 3D dynamics to train dynamics models with particle [24, 25], mesh [26, 7], or object [27, 28] representations, and these models can be more accurate than analytic simulators in some circumstances [8]. However, the requirement for training data with ground-truth 3D particle or object poses limits their applications. Works have attempted to train 3D dynamics models with perceptual data such as videos and point clouds, many leveraging Neural Radiance Fields (NeRF) [12]. One approach has been to apply point cloud distance functions such as Chamfer distance to define loss functions between predicted points and the next-step point cloud [29] or NeRF [30] observations, though these often require object segmentations and have not been applied for large-scale scenes. Another strategy pre-trains conditional or object-level NeRF renderers, then trains a dynamics model on these representations [31, 32]. This split training process and coarse representation is limiting for dealing with complex scenes and results in lower-quality predictions [9].

HD-VPD derives from Visual Particle Dynamics (VPD) [9], extended to support robot actions and larger, more detailed scenes. Relative to the broader literature, HD-VPD has less restrictive data requirements (2 RGB-D cameras) than other NeRF-based works or mesh- or object-based works, and it jointly predicts 3D dynamics and 2D video, unlike pure 2D video or point cloud models.

**Point cloud architectures** The rise of deep learning models has opened up new possibilities for processing and understanding 3D point cloud data. Models like PointNet++ [33], Point Transformer [34] and Point Cloud Transformers [35] have demonstrated strong performance in tasks such as object classification, segmentation, and registration. Similar to our work, RandLA-Net [18] focuses on building memory- and time-efficient architectures that scale up to large point clouds. However, these models are typically designed for static point clouds and do not directly model the dynamics of the scene. Our work introduces a new class of PCTs, called Interlacers, which are specifically designed for modeling the dynamics of large-scale point clouds. Performer-PCT architecture was introduced in [15], but applied in a different setting of scalable Robotics Transformers rather than dynamics modeling. In this paper, we show that intertwining Performer-PCT layers with the introduced here Neighbor-Attender layers is a key to achieving scalable and high-precision dynamics models.

# 6 Conclusion

We present a new high-density particle-based world model called HD-VPD and train it on real-world RGB-D and kinematic bi-manual robotics data. To support HD-VPD's high fidelity predictions we propose Interlacers, combining linear-attention techniques with GNN-inspired local neighborhood attention methods, can model point clouds of sizes 100K+, infeasible for previous PCTs and GNNs. HD-VPD outperforms previous SOTA particle-based world models both in prediction quality and computational cost. Finally, we show how HD-VPD can be used in robotic planning, as accurate world models for scenes involving objects and robotic agents operating on them.

# 7 Limitations

HD-VPD is a deterministic dynamics model, meaning that its predictions blur with long horizons and stochastic dynamics. We leave extending the model to probabilistic predictions and long-horizon tasks to future work. HD-VPD might be improved via (1) more detailed finger and arm modeling in the kinematic skeleton; (2) data with near misses for robot planning applications; (3) upweighting points most relevant to robot interaction in training loss computations; (4) using loss functions to enforce physically realistic 3D motion rather than only correct RGB values. Planning with visual costs might avoid any mismatch between particle motion and physical outcomes.

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

# Appendix

## A    Performer-PCTs linear attention: closer look

Let $\mathbf{X}_{\text{in}} \in \mathbb{R}^{N \times d}$ be an input to the Transformer's attention block, where $N$ stands for the number of points and $d$ is the dimensionality of their corresponding latent feature vectors. The output $\mathbf{X}_{\text{out}} \in \mathbb{R}^{N \times d}$ of the regular attention module can be computed as follows for learnable $\mathbf{W}_{\text{Q}}, \mathbf{W}_{\text{K}} \in \mathbb{R}^{d \times d_{\text{QK}}}$, $\mathbf{W}_{\text{V}} \in \mathbb{R}^{d \times d}$ and query/key dimensionality $d_{\text{QK}}$:

$$\mathbf{X}_{\text{out}} = \mathbf{D}^{-1} \mathbf{A} (\mathbf{X}_{\text{in}} \mathbf{W}_{\text{V}}), \ \mathbf{D} = \mathbf{A} \mathbf{1}_N$$
$$\mathbf{A} = \exp \left( \frac{(\mathbf{X}_{\text{in}} \mathbf{W}_{\text{Q}})(\mathbf{X}_{\text{in}} \mathbf{W}_{\text{K}})^{\top}}{\sqrt{d_{\text{QK}}}} \right) \tag{2}$$

In Eq. 2, $\mathbf{1}_N \in \mathbb{R}^N$ denotes the all-one vector and matrices: $\mathbf{X}_{\text{in}} \mathbf{W}_{\text{V}}, \mathbf{X}_{\text{in}} \mathbf{W}_{\text{Q}}, \mathbf{X}_{\text{in}} \mathbf{W}_{\text{K}}$ are often referred to as: $\mathbf{V}$ (value), $\mathbf{Q}$ (query) and $\mathbf{K}$ (key) respectively. Furthermore, mapping $\exp$ is applied element-wise. Finally, matrix $\mathbf{A}$ is called the *attention matrix* and encodes how points attend to each other via the so-called *softmax-kernel* (defined as: $\text{K}(\mathbf{u}, \mathbf{v}) = \exp(\mathbf{u} \mathbf{v}^{\top})$ for row-vectors $\mathbf{u}, \mathbf{v}$).

Time- and space-complexity of the regular attention mechanism is clearly quadratic in the number of points $N$, since attention-matrix $\mathbf{A}$ has $N^2$ entries. This makes it prohibitively expensive for massive point clouds. Thus Interlacers use instead linear attention mechanism introduced in the class of Transformers, called Performers [14]. Performers propose the following computational model of attention, where $\phi : \mathbb{R}^{d_{\text{QK}}} \to \mathbb{R}^m$ is applied row-wise and $m$ is a hyper-parameter:

$$\mathbf{X}_{\text{out}} = \mathbf{D}^{-1} \mathbf{Q}'((\mathbf{K}')^{\top}(\mathbf{X}_{\text{in}} \mathbf{W}_{\text{V}})),$$
$$\mathbf{D} = \mathbf{Q}((\mathbf{K}')^{\top} \mathbf{1}_N), \ \mathbf{Q}' = \phi(\mathbf{Q}), \ \mathbf{K}' = \phi(\mathbf{K}) \tag{3}$$

The brackets indicate the order of computations. Calculations in Eq. 3 can be performed in time linear in $N$ (and in the hyper-parameter $m$ that in practice we choose as $m \ll N$). System defined by Eq. 3 is obtained from the one given by Eq. 2, by replacing regular attention matrix $\mathbf{A}$ with a matrix $\mathbf{A}' \overset{\text{def}}{=} \mathbf{Q}'(\mathbf{K}')^{\top}$. Following [15], we use $\phi = \text{ReLU}$ applied element-wise (thus $m = d_{QK}$).

## B    Hardware Setup

Overview of the hardware setup is shown in Figure 11

**Cameras and Image Processing** We use a Realsense D415 overhead camera and Realsense D405 attached to the left wrist. We sample the overhead camera at 480x640 resolution, and the wrist camera is captured at 480x848 resolution and cropped to 480x640. We inpaint the overhead camera depth images via cv.INPAINT_TELEA. We mask both cameras observations to exclude points more than 2 meters from the cameras according to the depth images.

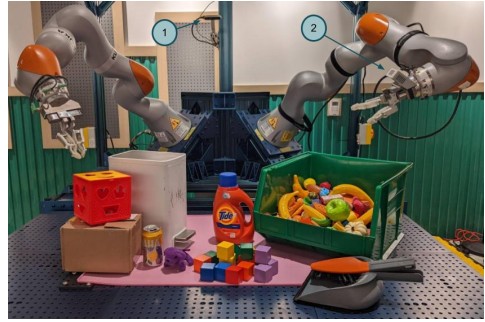

Figure 11: Hardware setup. The robot consists of two kuka iiwa arms extended over a table, and 2 realsense cameras (1. overhead, 2. left wrist). Data is collected from 5 of these cells with some variation in background and lighting. A representative sample of objects from our dataset is shown.

## C  Action Space

We represent actions as 26 particle kinematic skeletons at the desired future state positions. There is an additional particle representing where the fingertips would be if the gripper was closed, particles representing the estimated current position of the fingertips. Also, there is an additional particle on the side of each wrist so that the orientation of the gripper can be disambiguated especially when the fingers are closed. The fingers themselves are somewhat compliant so the skeleton points do not exactly reflect the true pose of the fingertips especially when in contact with the environment. The origin of the space is at the corner of the pink mat closest to the base of the right arm (no wrist camera). An example rendering of this skeleton is shown in Figure 12.

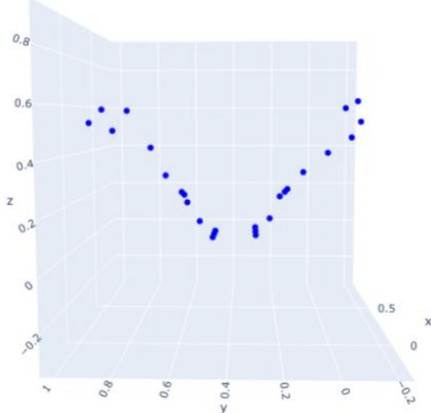

Figure 12: Action Space. Consists of 26 points representing a kinematic configuration of the robot.

## D  Grasp Experiments

For the grasp planning real world evaluations, two threads of fishing wire where attached to the bottom of the can and run through the table and then tied to a small weight. This way whenever the Coke can is dropped, it returns to exactly the same pose for all rollouts.

We start with an initial grasp centered at the center of the can, and then back off the grasps along the y-axis of the table (aligned with pink mat) in 5mm increments. Figure 10b overviews the relationship between these offsets and predicted plan cost. Figure 13 shows the observations from the left and overhead cameras snapshot during the real world rollout of these planned grasps.

## E  Training and architecture details

We train all models with a batch size of 16 for 1M total steps. We use a learning rate schedule of `3e-4` until step 1000, then `1e-4` until step 100K, then `3e-5` until the end of training. We use the AdamW optimizer [36] with weight decay `1e-3` and clip the gradient norm to `0.01` to prevent outliers in the dataset from destabilizing training.

All models are trained end-to-end with 6-step rollouts during training. At each of the two input timesteps and the 6 predicted timesteps, losses are computed on 128 sampled rays. Rays from input timesteps are rendered conditioned on the particles encoded from their respective input timesteps, while rays from predicted timesteps use the particles predicted by recursively rolling out the dynamics model in particle space conditioned on a sequence of actions.

The encoder uses a U-Net [16] with `[32, 64, 128, 256, 256, 128, 64, 32]` channels applied to each input image. It outputs 16-d per-pixel feature vectors. Points from the encoder are only included in the set of dynamics particles if they fall within the robot's workspace. Input and target RGB-D images are masked using the depth channel, changing the color of any pixel whose depth is > 2m to solid white.

Successful Grasp (No offset)

Time

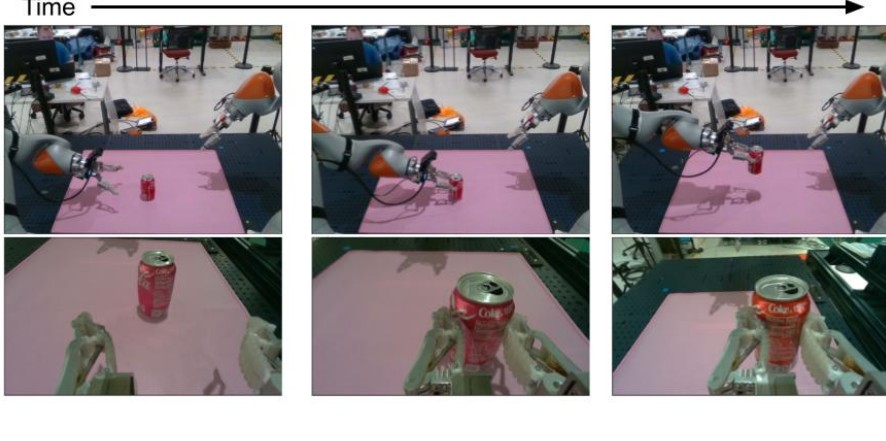

Failed Grasp (With offset)

Time

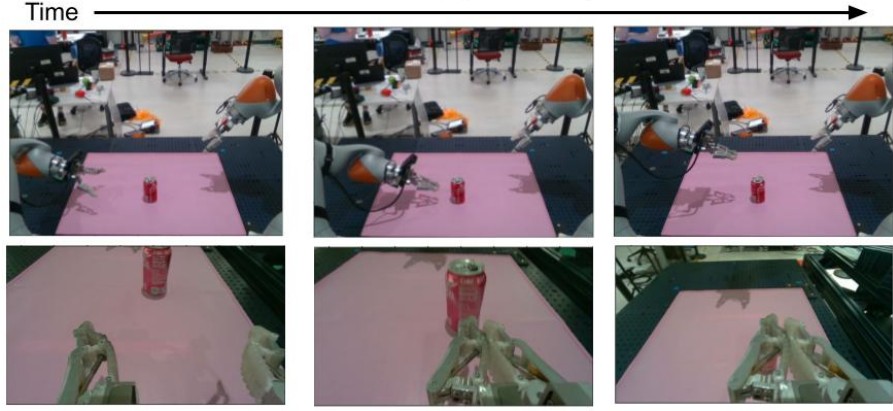

Figure 13: Two exemplar rollouts from the grasp experiments. The top grasp has no offset from the center of the can. The bottom grasp has an offset from the center of the can and misses.

The Interlacer dynamics model uses one Neighbor-Attender block and one Performer-PCT block on each input timestep of scene particles, using separate weights for each input timestep, and one (quadratic) PCT block on the kinematic particles. These three point clouds are then concatenated. Then this large, multi-timestep, multi-input-modality point cloud goes through another Neighbor-Attender block, followed by 3 Performer-ReLU blocks. All Performer-ReLU layers use a key dimension of 16 and a value dimension of 64. This dynamics model outputs $(\Delta\mathbf{x}_i, \Delta\mathbf{f}_i)$ for each particle $i$ of timestep $T - 1$. The $\Delta\mathbf{x}_i$ values are constrained to be in $[-0.15, 0.15]$ using a tanh and scaling, preventing any particle from moving more than 15cm in any single step. The output feature vectors $\mathbf{f}$ are 16-d to match the features coming from the encoder. To compute neighbors in the Neighbor-Attender layers, we use a CUDA kernel generated by Pallas (https://jax.readthedocs.io/en/latest/pallas/index.html). This same kernel is used for finding nearby points in the renderer.

The renderer follows the same scheme for constructing input features for the NeRF MLP as Whitney et al. [9]. We use four concentric annular kernels with radii [0, 0.01, 0.02, 0.05] and corresponding bandwidths [0.01, 0.01, 0.01, 0.05], and these kernels are approximated using 16 nearest neighbors. The near plane for ray rendering is set at 0.1m and the far plane at 2m. Rays are composited against a solid white background.

Reconstruction | Prediction

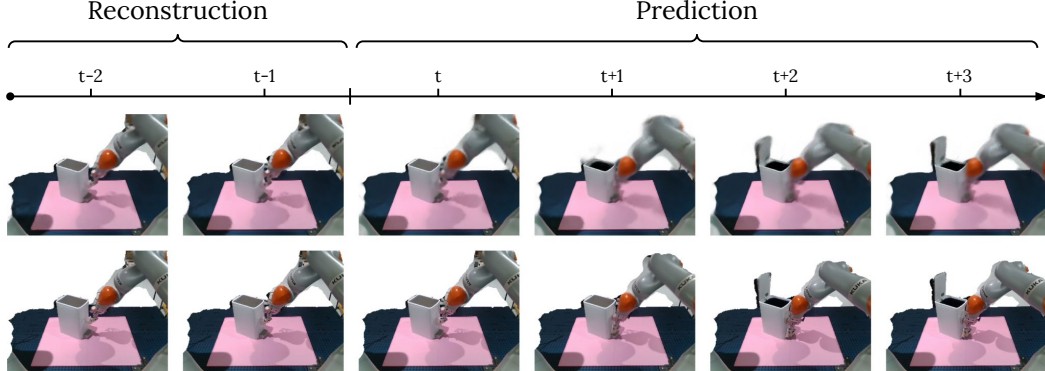

Figure 14: HD-VPD can accurately predict the dynamics of real-world interactions between robots (16-dof bimanual kukas) and complex objects, e.g., an articulated trash can whose lid opens upon pushing a pedal as shown here. This model trained on several hundred examples randomized over trash can pose, images here are from the test set. **Top row:** Renders from HD-VPD. Timesteps $t-2$ and $t-1$ are reconstructions of the two input timesteps; timesteps $t \ldots t+3$ are predictions into the future given robot actions. **Bottom row:** Ground-truth test set video.

## F   Dataset inspection

We introspect our training data by computing a histogram, shown in Figure 15, of HD-VPD's loss on a sample of the training set and using that to set a threshold for unusually high losses. Once we have set this threshold, we can search the training data for trajectories with high loss.

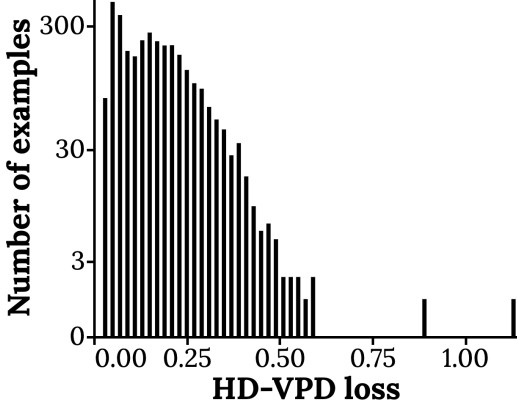

Figure 15: Loss histogram of HD-VPD on a sample of the training set. The data has a long tail of outliers with losses many times the mean. Note the log scale on the Y axis.

In Figures 16 and 17 we present examples of dataset quality issues discovered by inspecting training examples where HD-VPD's loss is anomalously high. These issues with our dataset were not previously known, and might pose problems for other applications of this data, such as policy learning. Discovering them with HD-VPD also allows us refine our hardware setup, fixing calibration issues and avoiding problematic robot states.

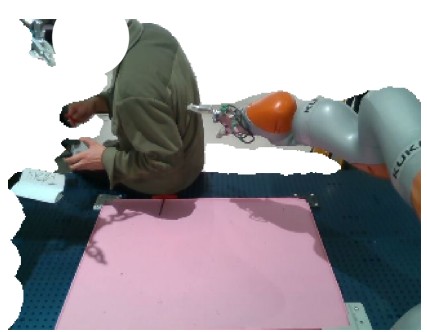 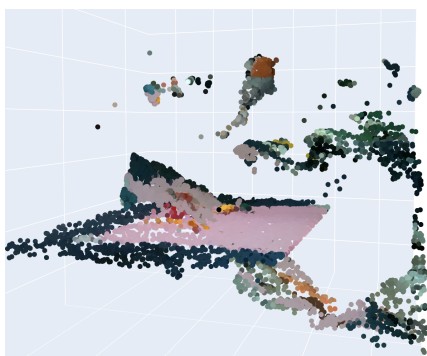

(a) Person in workspace     (b) Camera miscalibration

Figure 16: Two examples of problematic examples in our training dataset. **(a)**: A person is in the robot workspace. **(b)**: The wrist camera calibration is drastically incorrect. Visualized here are the points from the wrist and overhead cameras together; note the two intersecting planes of the table.

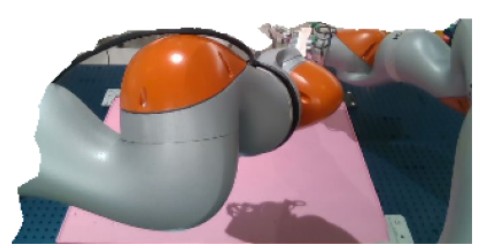 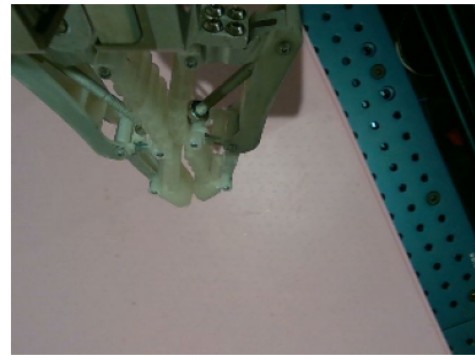

Figure 17: Another example of a problematic data point. The robot pose almost completely occludes the overhead camera's view of the scene, and the color calibration of the wrist and overhead cameras is very different.

