# OpenReview forum: "Modeling the Real World with High-Density Visual Particle Dynamics"
_robot-learning.org/CoRL/2024/Conference — CoRL 2024_

### Official Review · Reviewer_F4kU · 2024-06-28
**I like the ideas presented in this paper and the focus on efficiency and real-world applications.**

**Originality:** 3
**Technical Quality:** 3
**Clarity Of Presentation:** 3
**Potential Impact:** 3
**Recommendation:** 3
**Confidence:** 4

**Review:**

**Strenghts**:
* Novel architecture combining linear attention and neighborhood aggregation ideas from GNNs.
* Significant data collection effort.
* Real-world robotic application.
* Nice visualizations.
* Convincing performance gains over baselines.
* Computational complexity analysis.
* Detailed appendix.

**Weaknesses**:
* Visual reconstruction/prediction quality is low and results in very blurry generated videos.
* Writing and organization (Clarity): I found it quite hard to understand the downstream applications section – the heading is followed by several figures and then jumps to mechanically describe tasks and results without explaining the setup, e.g., “we demonstrate how HD-VPD can be used in real-world downstream tasks by using it to plan..” or something like that. I highly encourage the authors to improve the writing in this part.
* Downstream results: only graphs, no images/videos of successful/failed executions.

**Limitations**: Most of the limitations are clearly addressed. I encourage the authors to further discuss failure cases and an analysis that can provide insights into what parts of the model can be improved.

**Quality Of The Limitations Section:**

3

**Questions For Rebuttal:**

* Prediction window: throughout this work, the authors use l=2, meaning T-2, T-1 and T. To my understanding, the method observes 2 frames and performs a one-step prediction? Then, an autoregressive process is applied to predict more time-steps. Is it possible to train the model on one-step predictions but in-parallel? For example, Transformers usually train with a one-step prediction loss, but they are using “teacher-forcing” to train with a larger horizon (without autoregressive training, only at inference).
* Particle merging from different views: are you using a common method to merge different views? I couldn't find details on that in the appendix.
* Action representation: I find it interesting that you used one-hot vectors to encode the temporal position of the joints. In Transformers, it is common to learn additive embeddings, have you experimented with this kind of positional embedding?
* Other loss functions: as I mentioned, the prediction/reconstruction visual quality seems low. Have you considered other losses such as the pixel-wise L1 loss?
* Anchor particles: have you considered other sampling methods other than uniform, e.g. farthest-point-sampling (FPS)? What happens if uniform sampling yields two particles that are very close, won’t that be “wasteful”?
* Do the authors plan to release an open-source code for further research?

**Robotics Focus:**

4

**Summary Of Paper:**

This work proposes HD-VPD, a method combining a particle-based visual representation with an efficient Performer-based Point Cloud Transformer that utilizes GNN-inspired neighborhood aggregation, Interlacers, for dynamics modeling for physics and interaction prediction. The proposed dynamics module is more efficient than previous GNNs used for similar tasks and produces comparable or better performance on dynamics prediction. Notably, the method utilizes only two camera views and actions for training.

**Summary Of Recommendation:**

Overall, I like the ideas presented in this paper and the focus on efficiency and real-world applications. I appreciate the data-collection effort and I find several key contributions that are novel with moderate impact and significance. I have raised several concerns in the “Weaknesses” and “Questions” parts of my review that I would like the authors to address, but I think the paper is relevant and would be interesting to the CoRL community. As such, my current recommendation is “Weak Accept”, but I’m willing to increase my score given convincing answers to my concerns and the concerns that might be raised by the other reviewers. **Post-rebuttal**: I appreciate the authors' clarification regarding the merging process, I highlt recommend including it in the paper for the curious reader. I'm still not convinced regarding the autoregressive training of the Transforner. The authors claim that "Performing a few steps of rollout during training helps the model learn to correct its own errors and produce stable long predictions, and is especially helpful for a deterministic model.", which I think is a very strong claim without any ablation of the rollut steps (which is set to 6 across all experimetns if I understood correctly). I think there is a general consensus that the prediction quality is not great but the method is still applicable for downstream tasks. I thank the authors again, and given the above I'd like to keep my score.

---

### Official Review · Reviewer_L6Jk · 2024-07-20
**Not super impressive results and better presentation needed**

**Originality:** 3
**Technical Quality:** 3
**Clarity Of Presentation:** 3
**Potential Impact:** 3
**Recommendation:** 3
**Confidence:** 5

**Review:**

### Strength
1. Overall pipeline is reasonable. The problem is well motivated and the experiments show that more particles lead to better future video prediction, which is a nice result and justifies the motivation.
2. The experiments clearly show benefits over GNN in terms of the memory requirement.

### Weakness
1. Generated video quality is poor. Future prediction tends to become blurry even without contact. This puts the capability of the model in modeling fine-grained dynamics in question.
2. When comparing to GNN, the performance gain seems to shrink with scale. This puts a doubt on the scalability.
3. The method is somewhat incremental over VPD, with mostly an architecture change. Furthermore, it is unclear how the method compare with VPD from the experiments section. While I understand that VPD does not include the robot actions and may not scale well, it should be necessary to make clear other differences both in text and through experiments.
4. The application to control is not clearly demonstrated. If I understand correctly, the planning is done over simplified action space, such as the grasping offset or the pushing distance. Is the model not fast enough for planning more complex actions? Visualization in figure 8 and 9 are not clear enough; Would be nice to extend the x-axis of figure 10 (a)

**Quality Of The Limitations Section:**

2

**Questions For Rebuttal:**

Why are the generated future predictions blurry? The demonstrated case mostly deal with simple rigid body dynamics such as pushing and grasping, which should be fairly deterministic. Additionally, it seems that the generated videos become blurry even before the robot makes any contact.

**Robotics Focus:**

4

**Summary Of Paper:**

The paper presents a new model architecture which combines linear attention and GNN for training a 3D particle dynamics model with more particles

**Summary Of Recommendation:**

Major concern is the quality of the generated videos. This seems worse compared to prior work such as VPD

---

### Official Review · Reviewer_FVjU · 2024-07-30

**Originality:** 3
**Technical Quality:** 3
**Clarity Of Presentation:** 3
**Potential Impact:** 2
**Recommendation:** 3
**Confidence:** 3

**Review:**

### Strength
* The proposed method is more time and memory-efficient while achieving comparable or better performance compared with a previous GNN-based method for the robot dynamics prediction.
* The proposed model represents robot actions as a set of robot joint particles as input for the dynamics model. This overcomes the limit of VPD [9] which can only work on passive dynamic scenes (without actuation).
* The writing is clear and easy to follow. The Appendix and the website provide additional details and animations to promote an easier understanding of the paper.

### Issues
* The contribution on the technical side is a bit incremental. The overall framework is an improvement upon VPD [9] by replacing the GNN-based dynamics model and adding additional robot joints as input. The presented dynamics model, i.e., the Interlacer, combines a previous work Performer-PCT layer [14,15] and a neighbor-attenter, which is similar to the local feature aggregation in RandLA-Net [18].
* The proposed method is only trained and evaluated in a single setting. Since the method is not limited to bi-manual robots, it would be more convincing if the method could be evaluated in other public robot datasets, such as Droid dataset [a].
* Clarification: It's unclear how the dataset is split for training and testing. Since the dataset includes 5 different robots and 60 tasks,  I would like to see how generalizable the model is with different robots, unseen objects, and unseen tasks.
* What is the performance of the proposed Interlacer architecture on the Deformable Collision dataset [9]?

### Typos
* Line 75: "...the scene state and and..." (duplicated "and")

[a] Khazatsky, A., Pertsch, K., Nair, S., Balakrishna, A., Dasari, S., Karamcheti, S., ... & Finn, C. (2024). Droid: A large-scale in-the-wild robot manipulation dataset. arXiv preprint arXiv:2403.12945.

---
Post-rebuttal updates:

Thanks to the authors for the added evaluation of HD-VPD vs VPD on the deformable collision dataset. It's interesting that the proposed model performs slightly worse than VPD, on both plots and the GIF in the rebuttal material. This is somehow consistent with the observation (also from other reviewers) that the predicted video is a bit blurry.

It's also great to see the additional robot execution videos, which reduces my (and other reviewer's) concern on how to use it for real robot planning.

I also appreciate that the authors promise to conduct an evaluation on held-out data in the camera-ready version. They also promised to open-source the code.

Regarding DROID: I still think the proposed method can benefit from these modern large-scale robot datasets (e.g., droid, open x, rh20t, etc) and potentially generalize better for downstream applications, since these datasets contain more diverse scenes, robot motions, camera views, etc.

I changed my rating of `Robotics Focus` and increased the final score.

**Quality Of The Limitations Section:**

2

**Questions For Rebuttal:**

I think the following things could further improve the paper:
* Clarify the training/testing protocol in experiments.
* Benchmark the model on the Deformable Collision dataset [9]
* Add experiments for unseen robots, objects, and tasks.
* Add experiments on public robot manipulation datasets such as droid.
* Varying the number of cameras.

**Robotics Focus:**

4

**Summary Of Paper:**

This manuscript presents a method to predict visual particle dynamics of the real-world interactions between robots and robots and objects given an input scene and the robot actions. Specifically, a new point cloud transformer called Interlacers is introduced. This architecture intertwines efficient linear-attention Performer layers proposed in [19] and graph-based neighbor attention layers. Experiments were conducted on a bi-manual Kuka arms setting with two cameras. Compared to the previous Graph neural network-based method, the proposed method achieves comparable performance with less computation time or achieves better performance using 4x more particles. Experiments also demonstrate the potential of using this model for downstream tasks, such as rollout evaluation for robot planning and detecting outliers from a training dataset.

**Summary Of Recommendation:**

The paper presents a new architecture that can model the visual particle dynamics of a robot manipulation scene given future robot actions. Although the new architecture is more efficient compared to prior work VPD, I feel the technical novelty in this architecture is limited and some additional experiments should be conducted on the Deformable Collision dataset. The experiments are missing important details and the generalization ability is unclear. The training is limited to one robotic dataset and the paper can be further improved by adding more experiments on public robotic manipulation datasets such as Droid.

---

### Author Rebuttal · Authors · 2024-08-13

This file contains results including plots and videos on the Deformable Collision dataset, along with videos of successful and failed planned grasps executed on the real robot.

---

### Decision · Program_Chairs · 2024-09-04

**Decision:**

Accept

**Comment:**

**Post-Rebuttal Guidance**.
Guidance: accept, but borderline. This is clearly a weak accept/borderline paper - If the PCs have limited capacity on accepting papers and would like to reject this paper, I would not argue for its acceptance. Reading the rebuttal between the authors and the reviewers, I suspect there is some hesitancy in accepting the results of this paper. First of all, this approach is only validated on a single robot and single dataset - initial results were provided on the Deformable Collision dataset (but the proposed approach had comparable performance with the baseline). All three reviewers admitted that the prediction results were not high quality. While I do appreciate that the proposed approach is more time/memory efficient, I feel that it is incomplete without demonstrating better performance on more datasets (e.g. Droid) or on more tasks.

**Pre-Rebuttal Summary**

Strengths:
- The proposed approach is more time and memory efficient with comparable or better perf
- Clear writing

Weaknesses:
- Limited novelty over prior work (VPD, Performer-PCT, etc.)
- Evaluation is not as convincing - only 1 setting is shown. Reviewers have suggestions for other datasets (eg. Droid)
- Generated reconstruction quality is still very poor

The reviewers have mixed responses to the paper - authors hope you could check and address them